# Determinants of Tourism Product Development in Southeast Ethiopia: Marketing Perspectives

**Kassegn Berhanu Melese** [1],* and **Temesgen Heiyo Belda** [2]

1 Department of Tourism Management, Debre Berhan University, Debre Berhan P.O. Box 445, Ethiopia
2 Department of Tourism Management, Assosa University, Assosa P.O. Box 18, Ethiopia; www.ttemesgen.com@gmail.com
* Correspondence: kassegnberhanu@gmail.com

**Abstract:** Tourism has been given much attention in developing countries like Ethiopia. In this regard, tourism product development played a great role in achieving sustainable developmental goals. The study aims to examine the determinants of tourism product development in southeast Ethiopia. The research employed a mixed research approach, and descriptive and explanatory research design was used. Both secondary and primary data sources were in place to obtain the relevant data. A total of 398 samples were employed to collect the data. A convenience sampling technique was employed to select domestic tourists and walk-in guests and purposive sampling was applied to select marketing managers of hotels, lodges, boat associations, park administration, resorts, restaurants, nightclubs, and guest houses. The qualitative data was analyzed through thematic analysis and the quantitative data was analyzed using descriptive and inferential statistics by computing SPSS. The study revealed that tourism product development is affected by marketing mixes of which promotion was the major factor of tourism product development and price was the only tool inversely related to tourism product development. The brand image also positively determined tourism product development. The results showed that the exercise of tourism product development strategies among tourism businesses was low. The main implication drawn from the study is that the tourism businesses have to develop new customer profiles or segmenting in customers in their specified characteristics like gender, level of income, and age. Furthermore, the need to invest in research and development of the current market to develop new tourism products is of great importance.

**Keywords:** tourism; Ethiopia; product development; marketing mixes; brand image; determinants; sustainability

## 1. Introduction

Tourism activity is a combination of the use of tourism products like transportation, accommodation, infrastructure, attractions, and support services. These products highly influence the demand for tourism at the destinations [1]. According to Khan, Hassan, Fahad, and Naushad [2] economic development and tourism development have a direct relationship. In this regard, tourism product availability, product quality, and product management played a great role in developing the tourism industry [3].

Tourism product development, in tourist receiving destinations, highly influences the development of the tourism industry [4]. In this article the authors discussed that the destination may be rich in tourism resources but the growth of travel and tourism would depend on tourism product development. Tourism product development is determined by marketing, political, environmental, technological, social, and economic factors [5].

According to Camilleri [6], the function of tourism is serving tourism products for travelers from their homes to their destinations. Tourism product can be defined as the conjugation of visible and invisible components, i.e., cultural, human-made attractions, service and facilities which create travelers' experience and motivation to potential customers and

it is valued at price and ready to sell for actual travelers and prospects [7]. According to the 2016 report of Central Board of Secondary Education cited in [8], Western nations such as France and Switzerland have accumulated the highest share of economic and social welfare from the tourism industry. As stated by Worku and Tessema [9], tourism in developing countries including Sub-Saharan countries has shown fast growth. As per the report of World Travel and Tourism Council [10], the Ethiopian tourism sector has registered the greatest rate in the world, which is 48.6%.

Moreover, the country is rich in cultural heritages which show the past and the current generation's manifestation and identity [11]. However, tourism product development is limited to some areas even within the same region due to natural and historical endowments of potential tourism product distribution, hence resulting in unbalanced growth.

The tourism development potential in Ethiopia is still hindered by many factors [12]. The authors found that lack of skilled human power, lack of tourist information centers, less marketing and promotional activities, inadequate infrastructure, lack of tourism organizations travel agents, tour guides, and tour operators are the major challenges for tourism development in the country. In 2017, the number of foreign tourists who visited Ethiopia was about 871,000 [10].

Strong control and effective management of the main tourism products like attractions (natural, historical, cultural, paleontological, and archaeological), accessibilities, amenities, and ancillaries leads to tourism product development and lack of coordinating these tourism product components leads to failing in tourism development [13]. The growth of per capita income and the exchange rate have a direct relationship with tourist supplies [14]. However, the studies focused on tourism development and tourism product development and therefore showed a gap in investigating the determinants of tourism product development from a marketing perspective at the site level. Further, many of the studies were conducted in developed nations' contexts making the generalizability of the findings difficult for developing countries like Ethiopia. Akama and Kieti [15] discussed how tourism product development affects tourist attitude and buying behavior. However, the study did not address the determinants of tourism product development. Moreover, Fiseha, [13] has found that tourism product development is assured when all tourism resources are developed together but his study failed to identify factors that affect tourism product development. Furthermore, Chiriko [16] found that cultural tourism product marketability in Sidama Region, Ethiopia is under low marketable value and is highly dependent on the typology of tourist arrival.

Therefore, little is known about the determinants of tourism product development in developing countries like Ethiopia at the destination level. All the above-related studies tried to emphasize tourism products from different dimensions but none of them studied marketing determination of tourism product development in the studied geographical area. Factors that determine tourism product development in Southeast Ethiopia remain one of the overlooked research areas; hence, this study aims to fill the gap through examining the determinants of tourism product development in Ethiopia, taking Batu town and its vicinities as a study context. To this end, the study will address the following objectives:

1. Explaining tourism product development strategies used among tourism businesses.
2. Examining the effect of marketing mix elements on tourism product development.
3. Investigating the effect of the brand image of tourism service providers on tourism product development.

## 2. Literature Review Graduate

### 2.1. The Concept of Tourism

There is no universally accepted definition for tourism. Theobald [17] defined tourism as a "tour" which is derived from the Latin word "tornare". In the Greek language, "tornos" means circle rotating from the central point. Nowadays, the meaning is "once turn" and the suffix tourism (ism) stands for action for travel or the process needed to travel. Therefore,

when the words "tour" and "ism" are combined, it will give the meaning of action to travel around, over the world or circle.

The word tourism is an activity of traveling to somewhere and staying more than a night away from one's resident place (home), not for more than a year for different purposes (business, education, leisure, etc.), and not participating in remunerative activities within the visited place.

These days, tourism is experienced by many people around the world. Especially the last two decades were a crucial time for the development of the tourism sector because of technological changes and globalization effect. Tourism is a unique common denominator that combines different industries and service providers like parks, restaurants, the aviation sector, transportation, accommodations, and museums.

### 2.2. Tourism Product Development

Product development starts from idea generation, business analysis, testing the market, and commercializing. Therefore, tourism product development is the completion of all the life cycle of the product to meet the customers' needs: the standard they need to obtain and the entire journey of the product itself considered and expressed as tourism product development.

As narrated by Alemshet [1], tourism product is the aggregate of perception and presentation of attraction and transportation facilities accompanied by standard services. Hence, the indicators of tourism product development include attractions (i.e., natural, cultural, and archeological) and transportation services which are suitable for the attraction sites with acceptable and suitable accommodation services, accurate and timely information, and desirable food and beverage products. Therefore, the author added that tourism product development attracts and increases the flow of tourists to the area and highly raises the revenue of tourism service providers and the diversification of tourism product.

Tourism product development is defined as either creating new products injected into the market or innovating and rejuvenating the existing products, experiences, and services as per the tourists' needs and wants [18]. It is argued that either creating or improving the product may attract different types of visitors and the situation upgrade sales of tourism products, strengthen the market positioning, or expand new market opportunities.

### 2.3. Marketing in Tourism and Tourism Marketing Mixes

The main aim of tourism marketing is the production and placement of products to the targeted customers [19]. According to the authors, tourism marketing helps to gather and analyze information about the actual and future offer of goods and services to the targeted users. After developing tourism products, the next issue is marketing and promotional activities which lead the customers to purchase the product [1].

Marketing mixes are variables that are controlled by the firm or organization to satisfy the targeted customers [20]. The authors narrated that marketing mix was explained in 1960 by Borden and Culibon cited in [20], and it has 4Ps: price, place, promotion, and products. These help in the production and creation of a transaction to deliver and satisfy the target customers [21]. Kotler and Armstrong [22] defined marketing mixes as controllable variables that have the power to change the buying attitude of the customers. In this respect, the oldest 4Ps of marketing are questioned as they missed some points. As a result, it has been extended into 7Ps by including physical evidence, people, and process. These 3Ps are used further in the case of service provider organizations and industry. Therefore, since tourism is more of a service-oriented and independent service industry, the researchers use the 7Ps of the marketing mix to examine the determinants of tourism product development in the study area.

#### 2.3.1. Product in Tourism

A product is anything offered to the market that may directly or indirectly be used or consumed for the satisfaction of the users [23]. The product can be expressed in terms of per-

sons, places, goods, services, ideas, and organizations. Panizzon, Vidor, and Camargo [24] coined that a product is complex and might be a visible or invisible attribute, including packaging, color, price, prestige, and services that create the experience by satisfying customers' needs. Therefore, from the above illustrations, product is a collection of physical and psychological elements that are needed by the market and to fill the expectation of the customer. Moreover, the product is considered as the basic and the first component of the marketing mix. Developing the existing product is considered as the destabilization of the market and the new product development is considered as the chaos of the market [25]. Mason and Staude [25] concluded that developing products is a fundamental and tactical strategy for competitors from the market.

World Tourism Organization [26] argued that the demand point of view (visitors' perspective) is the perception of customers for their expenditure which includes package tours, food and beverage, accommodation services, transportation, recreational activity, and shopping. The supply point of view is the analysis of the production process by suppliers and it includes consuming and non-consuming products.

Madafuri [27] contemplated that three elements make tourism a product: attraction and its environment, facilities, and accessibility.

Tourism product is a process that used the destination resources to meet the needs and wants of both domestic and international tourists [28]. This study clearly shows that developed countries consider tourism as a major sector always characterized by diversified tourism products and well-developed tourism products and developing countries are very committed to developing the tourism product. However, under-developed countries have to focus on limited or small-scale development. Based on the above empirical reviews, the following hypothesis is proposed.

**Hypothesis (H1):** *There is a statistically significant and positive relationship between the availability of tourism products and tourism product development.*

### 2.3.2. Price in Tourism

Price is the total payment of the product consumed or purchased by the customer [22]. Pricing is also one of the marketing mix strategies which have the power of stabilizing or destabilizing the market. Eavani and Nazari [29] argued that price is the determinant of the level of satisfaction and the amount of currency that the product would be offered.

Cirikovic [19] underlined that high quality of tourism product needs high cost and the perception of customers for high pricing product is high quality. Therefore, pricing techniques and pricing management play a great role in winning the competition. The above literature leads to the development of the following hypothesis.

**Hypothesis (H2):** *There is a statistically significant and positive relationship between tourism price and tourism product development.*

### 2.3.3. Place in Tourism

Distribution and accessibility of products (goods and services) are considered as the basic stabilizing mechanisms of the marketing mix [25]. Managing the supply chain from supplier to producers and from producer to customers can create a successful distribution channel (place). According to Eavani and Nazari [29], distribution is the system of how to contact and deliver to the target customers by analyzing and solving issues like channels to distribution, target market area coverage, inventories, and means of transportation.

A tourism distribution system is a way of making tourism products (goods/services) available and accessible for the targeted travelers [19]. Place in tourism is about the provision of different tourist spots, information centers for different travelers, selecting tourism attractions and destinations [22]. Hence, the following hypothesis is established.

**Hypothesis (H3):** *There is a statistically significant and positive relationship between place (distribution) and tourism product development.*

### 2.3.4. Promotion in Tourism

According to Mason and Staude [25], promotion is mostly stabilizing and slightly destabilizing the market. Communication between producers and customers is very essential [30]. Effective communications about product design, price, structures, and features should have been well addressed to the customers via different promotional mixes like sales promotion, personal selling, advertising, public relation, and direct marketing tools [22,29].

Tourism product promotion is creating awareness and image for positioning in the market to attract potential tourists to the destination [19]. Based on the existing scholars' work, an assumption is proposed.

**Hypothesis (H4):** *There is a statistically significant and positive relationship between promotion and tourism product development.*

### 2.3.5. People in the Tourism

In the hospitality sector, the quality of people (behavior, quality control, personnel selling) is needed [22]. People create soul in the tourism industry and key stakeholders such as employees in the industry serve tourism product [31]. People who are engaged in tourism sectors should have a good attitude, educational background, skill performance, grooming, or the total appearance to create satisfaction and memorable tourist experience. People in the hospitality and tourism industry are mandatory because the service industry is intangible and may be expressed and available for customers via the presentation of the employee. Having analyzed the central values of people in the tourism and hospitality industry, the following hypothesis is developed.

**Hypothesis (H5):** *There is a statistically significant and positive relationship between people and tourism product development.*

### 2.3.6. Physical Evidence in Tourism

The physical evidence includes physically touched facilities like buildings, decors, gardens, parking lots, lighting systems, and colors, surrounding environments and everything which could be visible and attractive based on some standards during the use of tourism products by the tourists [32]. In their work, they elaborate that the physical evidence makes the intangible characteristics of services tangible and supports the tourism product development). Then the researchers develop a research hypothesis.

**Hypothesis (H6):** *There is a statistically significant and positive relationship between physical evidence and tourism product development.*

### 2.3.7. Process in Tourism

The process is a way of serving customers and relies on planning, anticipating, consumption and recollection, standard product delivery, time, and money [33]. The authors add that the standard service delivery period plays its role in tourism product development. As a result, researchers propose a research hypothesis as follows.

**Hypothesis (H7):** *There is a statistically significant and positive relationship between process and tourism product development.*

### 2.4. Tourism Product Providers Brand IMAGE

According to Sonnleitner [34], image is a very important aspect compared with other competitors' resources to attract potential customers. Further, he defines tourism destination reputation as the total reaction of the organization to the needs and expectations of individuals. Having reviewed scholarly works, the researchers established the following research hypothesis.

**Hypothesis (H8):** *There is a statistically significant and positive relationship between tourism product providers' brand image and tourism product development.*

*2.5. Tourism Product Development Strategies*

Tourism marketing strategy includes mechanisms and activities that are prepared by destination management and concerned bodies for utilization and maximizing marketing values of products [22]. They include that tourism marketing strategies encompass market penetration strategy, new tourism product development strategy, market development strategy, and product diversification.

## 3. Methods and Materials

*3.1. Description of the Study Area*

Batu, a town laying on the shore of Lake Ziway Ethiopia, is located 162 km south of Addis Ababa on the road connecting Addis Ababa to Hawassa in the East Shewa Zone of the Oromia Region of Ethiopia. Batu has a latitude and longitude of 7°56′ N 38°43′ E with an elevation of 1643 m above sea level [35] (see Figure 1: Map of the study area).

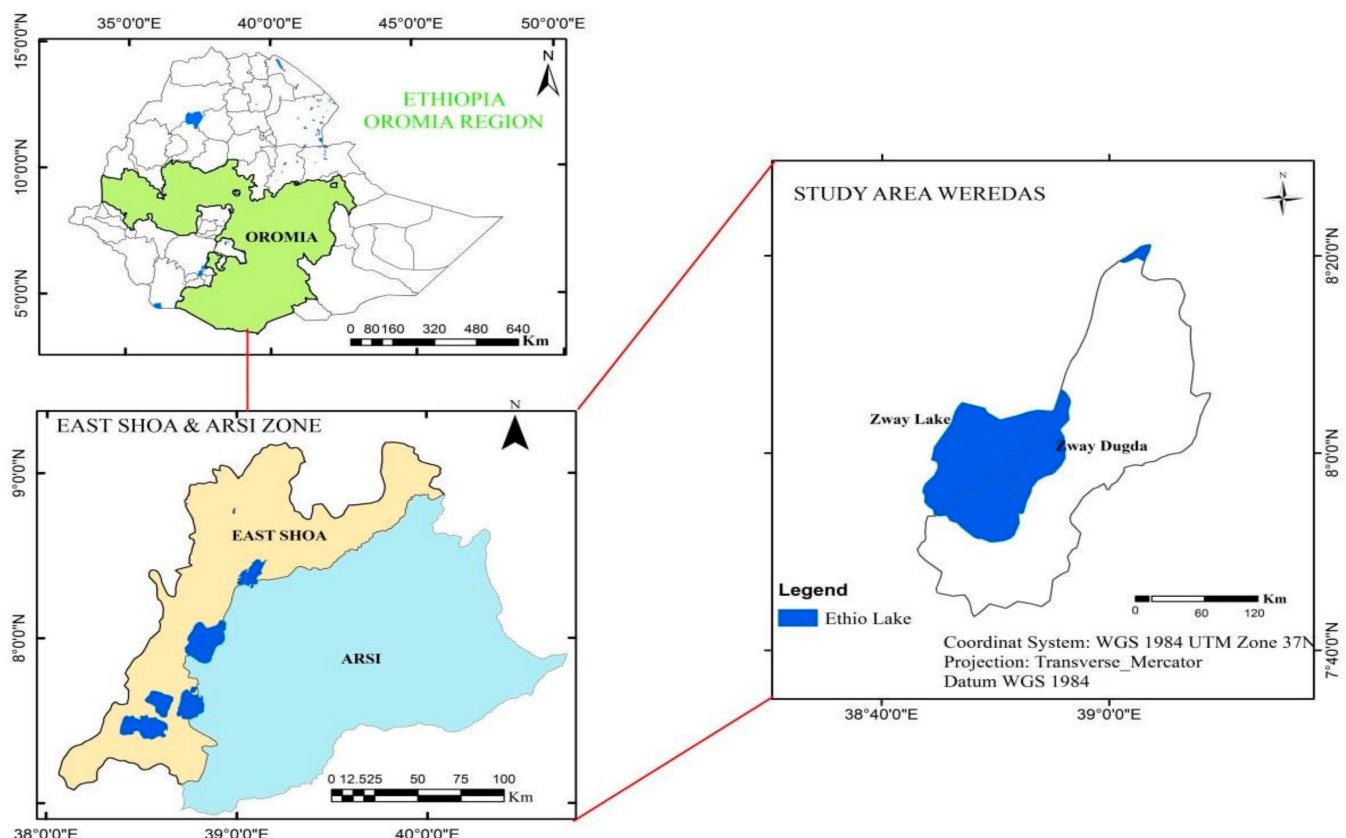

**Figure 1.** Map of the study area. Source: developed by authors, 2021.

Bird watching, horse riding, religious sites, geological sites (Lake), trips to Abijatta Shalla National Park, wildlife watching, and boat trips on the lake to visit the island monasteries where the Ark of the Covenant was residing for 800 years, parchments of historical bibles, and other national heritages are the main tourism products and activities practiced in the area [36].

The town and the surroundings are rich with tourism potential resources like attractions including Abidjata & Shalla Lakes and the Park is rich in flora and fauna, like birds (455 specious) grant gazelle, greater kudu, and black-backed jackals, Ziway lake, Langano lake, beaches, cultural resources like monasteries Tullu Guddo or Debre Tsion, and Galilla, Debre Sina, Funduro, and Tsedecha are the volcanic islands located in Ziway

lake [37]. However, the potential of tourism resources and products in the area are not well developed because of many factors including marketing of those products. There is a lot of biodiversity (flora and fauna) in Batu and its surroundings. The *acacia* trees are densely grown in the Batu vicinities which help for research and study of plants and attract scientists and tourists to the area [4].

### 3.2. Research Design

In this study, descriptive and explanatory research designs were employed. The descriptive research design was used for describing the existing situation or phenomenon of the study in detail [38]. Therefore, it was employed to examine the strategies used by tourism businesses in the study area to develop tourism products.

The explanatory research method is applied to examine the cause-and-effect relationship between variables [39]. Therefore, the current study also employed an explanatory research design to explain the effect of marketing mixes on tourism products development as well as how the brand image of tourism business organizations influences tourism product development.

A qualitative approach was used to narrate and interpret qualitative responses from interviews respondents. A quantitative approach was in place to generalize the findings resulting from descriptive and inferential statistics. The data was collected in Batu town and surrounding tourism product providers. Primary data and secondary data were collected from September 2020 to July 2021 through a questionnaire survey and interview. The questions or units of measurement (items) were prepared according to the model specification (see the questionnaire in Appendix A).

### 3.3. Target Population

The providers of tourism products and domestic tourists (customers) were the subject of this study. Tourism product providers (hotels, restaurants, souvenir shops, boat associations, parks, lodges, night clubs, and tourism offices) were targeted. The exclusive criterion for international tourists was considered as it was difficult to find international tourists during data collection because of travel restrictions and challenges caused by the COVID-19 pandemic. Hence, the study targeted domestic tourists and walk-in guests to obtain the relevant data.

### 3.4. Sampling Techniques and Sample Size Determination

The researchers used non-probability sampling techniques. Convenience sampling was employed to select domestic tourists and walk-in guests and purposive sampling was also appropriate for the selection head of marketing or general managers of tourism product suppliers.

Cochran [40] created the formula that is used for a large and unknown population in order to take appropriate sample sizes. The formula considers 95% level of confidence and 5% or 0.05 level of estimated precision. The sample size is determined using the following formulae. $n = \frac{p(1-p)z^2}{E^2}$, where n = is the sample size; $Z^2 = 1.96$ is the desired confidence level for 95%; $p = 0.5$ is the estimated proportion of an attribute that is present in the population; $E = 0.05$ is the desired level of precision, 5%. Hence, n = $(1.96)^2(0.5)$ $(0.5)/(0.05)^2 = 384$ sample would be employed to collect data from domestic tourists and walk-in guests via questionnaire using convenience/accidental selection technique. For qualitative data gathering, the study followed the concept of data saturation point. Data saturation point is when the researchers get similar answers from the interviewee; then the data is saturated and the researcher should stop collecting further interview data from respondents [41]. Accordingly, in this study, the interview data was saturated at the point of the 10th interviewee. Therefore, the researchers select mostly available tourism business organizations for the interview (marketing manager, general manager, coordinators, etc.) of resort hotels, national parks, business associations, cultural restaurants, beach and lodge, souvenir shop, guest house, tourist hotel, and night clubs. Generally, the study

involves 10 interviewees and 384 participants for the questionnaire survey making a total of 394 respondents for both questionnaires and interviews.

### 3.5. Variables and Model Specification

Qualitative data was narrated through content and thematic analysis whereas quantitative data was interpreted by using descriptive and inferential statistics (multiple regression and correlation analysis). Multiple regression model analysis is used for quantitative data analysis and helps to predict the quantitative dependent variable [41].

$$Y = \beta 0 + \beta 1 X 1 + \beta 2 X 2 + \ldots \beta k X k + \acute{\varepsilon}$$

$$Y = \beta_0 + \beta_1 X_1 + \beta_2 X_2 + \beta_3 X_3 + \beta_4 X_4 + \beta_5 X_5 + \beta x_6 \beta_6 + \beta_7 X_7 + \beta_8 X_8 + \acute{\varepsilon} \ldots \text{model specification}$$

Y = dependent variable, $X_1$, $X_2$, $X_3$ = independent variables, $\beta_0$ = Y-intercept (constant term), $\beta_1$, $\beta_2$, $\beta_3$, . . . = Slop coefficient for each independent variables, $\acute{\varepsilon}$ = the model error.

In this study "Y" is the dependent variable which is the tourism product development and $X_1$ = product, $X_2$ = price, $X_3$ = distribution channel, $X_4$ = promotion, $X_5$ = people/employee, $X_6$ = process, and $X_7$ = physical evidence. (See the marketing mix model in Table 1)

Mason and Staude [25] suggested a marketing mix model focused on marketing strategy for the complex and turbulent environment by using marketing mixes in South Africa. The current study adopts this marketing mix model because of the complex nature of the tourism industry. Since the tourism industry is broad in concept and interrelated with many industries, the tourism product development would be determined by many dimensions and the researchers look at those determinants from a marketing perspective only.

**Table 1.** Marketing mix model.

| Tourism *product* | |
|---|---|
| New tourism product development planning | New tourism product development involves customers in the process |
| Product innovation | Modifying the existing tourism product |
| Product customization | Local tourism product development for the local users and customization of the products for all users too |
| Product design/flexibility | Updated and latest design use |
| **Price of tourism product** | |
| Credit terms | Postpaid service for customers |
| Price leadership | Price setter of the tourism product because of dominancy in the market |
| Value for money | Customers get equivalent value from the product at a given price |
| Discount | Selling of product with a minimum price to attract and retain customers |
| Distribution of tourism product | |
| Changes in the channel | If needed substitute the current channels by other best alternatives |
| Intermediaries | Reduce No. of intermediaries' chain system or contact users directly |
| Reduce perishability | Selling to the maximum effort of tourism perishable |
| **Promotion of tourism product** | |
| Media | Changing the mind of customers or changing perceptions of the destination by using appropriate media advertisement. |
| Personal selling | Selling or booking tourism products face-to-face is especially helpful for awareness of new tourism products |
| Public relations | Public relations increase trust in the tourism product in the destination |

**Table 1.** *Cont.*

| | |
|---|---|
| Sales promotions | Appreciation purchase of the tourism product by giving some unusual advantages to the consumer |
| Word of mouth | Customers' recommendation to influence their friends and relatives by internet and orally |
| **People in *tourism and hospitality*** | |
| Training | Periodical training for updating employees with current job policies |
| Certified | Graduated with the right certification for the right position |
| Satisfied | Create a comfortable working environment and payment |
| **Process in *tourism and hospitality products*** | |
| Standard products | Produce the same products all the time |
| Timing | Deliver the requested product within standard time specification |
| Money savor | Saves customers unnecessary payment |
| **Physical evidence in tourism and hospitality services and products** | |
| Surrounding environment | The surrounding environment should be attractive |
| Design | Building and working area designated according to needed standard |
| Décor | Offices, working environments, lobbies depict decor well |

Source: adapted from [25].

### 3.6. Validity and Reliability

According to Kothari [38], validity is the items or constructs in the questionnaires that can measure the issues or contents of the study appropriately, which is about using a true measurement of the study findings. In this regard, the research design, the questionnaire, and interview questions and all the contents included in the research were developed from related scholars' work and adopted model specifications then evaluated by tourism experts.

Reliability is one of the crucial concepts which refer to how the results are real when the variables are measured at a different time with similar instruments and conditions [42]. Therefore, the questionnaire reliability was calculated statistically by Cronbach's alpha. Cronbach's alpha reliability coefficient normally ranges between 0 and 1. The closer Cronbach's alpha coefficient is to 1.0, the greater the internal consistency of the items in the scale. The size of alpha is determined by both the number of items in the scale and the mean inter-item correlations. The reliability test showed that all the variables had a Cronbach's value of greater than 0.93 (rule of thumb) in this research, which is commendable and deemed to be excellent (see the reliability tests or results of variables in Table 2).

### 3.7. Ethical Consideration

The researchers believe they have chosen the right participants for this study. Consequently, diverse ethical issues were taken into consideration during this research, from the administration of the research instrument with respondents to the acknowledgment of all the primary and secondary sources being used. Before collecting the data questionnaire and interview questions, the checklists were tested by experts and academicians who have knowledge in the area and later the questions were corrected based on the inputs received from the professionals. Moreover, the researchers highly care about the privacy of respondents and selected organizations for data surveys as promised during data collection.

**Table 2.** Cranach's Alpha Test.

| Items | Cranach's Alpha | Number of Items |
|---|---|---|
| Product | 0.955 | 4 |
| Process | 0.964 | 3 |
| Brand image | 0.928 | 3 |
| People | 0.958 | 5 |
| Price | 0.955 | 4 |
| Physical evidence | 0.939 | 3 |
| Place | 0.936 | 5 |
| Promotion | 0.971 | 5 |
| Product development | 0.950 | 5 |

Source: Authors survey, 2021.

## 4. Results and Discussions

### 4.1. Characteristics of Respondents

The total questionnaires that have been distributed were 384 of which 372 (96.8% response rate) were valid for analysis. A detailed characteristic of respondents is illustrated as follows.

From Table 3, a total of 274 (73.7%) were male and 98 (26.3) were female. A detailed general profile of participants is illustrated below in the Table 3. Demographic characteristics of interviewees are presented in Table 4.

**Table 3.** Demographic characteristics of respondents.

| | Category | Frequency | Percentage (%) |
|---|---|---|---|
| Sex | Male | 274 | 73.7 |
| | Female | 98 | 26.3 |
| Age | 18–29 | 256 | 68.8 |
| | 30–45 | 114 | 30.6 |
| | 46–60 | 2 | 0.5 |
| | Total | 372 | 100 |
| Marital status | Unmarried | 219 | 58.9 |
| | Married | 143 | 38.4 |
| | Divorced | 5 | 1.3 |
| | Widowed | 5 | 1.3 |
| Level Education | TVET certificate | 91 | 24.5 |
| | Diploma | 124 | 33.3 |
| | Degree | 128 | 34.4 |
| | Master's degree | 16 | 4.3 |
| | Other | | 13 |
| Years of loyalty | 5–1 | 298 | 80.1 |
| | 10–6 | 66 | 17.7 |
| | 15–11 | 5 | 1.3 |
| | 16–20 | 1 | 0.3 |
| | <1 year | 2 | 0.5 |
| | Total | 372 | 100 |

**Table 4.** Demographic characteristics of interview informants.

| Interview Code | Age | Position | Sex | Experience | Location/Area of Work |
|---|---|---|---|---|---|
| MMH | 33 | Marketing manager | M | 2 years | Resort "A" |
| PAN | 37 | Park manager | M | 3 years | National park |
| BAC | 32 | Coordinator | M | 5 years | Boat association |
| CRM | 51 | Manager | M | 1 year | Cultural restaurant |
| LBLM | 31 | Manager | M | 2 years | Beach and lodge |
| TSS | 30 | Owner | M | 2 years | Souvenir shop |
| OGG | 38 | Owner | M | 5 years | Guest hose |
| AVM | 33 | Manager | F | 2 years | Resort lodge |
| HGM | 34 | Manager | M | 2 years | Hotel |
| LVNC | 35 | Coordinators | M | 1 year | Night club |

Source: authors survey, 2021.

### 4.2. Descriptive Statistics of Variables

From Table 5, the findings revealed that price (mean = 2.82, SD = 0.81), place ( mean = 2.54, SD = 0.88), and people (mean = 2.44, SD = 0.93) have scored above group mean, whereas product (mean = 2.18, SD = 0.83), promotion (mean = 2.25, SD = 0.87), image (mean = 2.29, SD = 0.93) physical evidence (mean = 2.37, SD = 0.95), and process (mean = 2.21, SD = 0.75) have scored below group mean value of the variables. The mean value of dependent and independent scores approached "Disagree" labels. However, the range difference between the mean value of price and tourism product development was the highest in comparison to the other independent. The reason behind this was the significant but inverse relationship of price and tourism product development.

**Table 5.** Mean ratings of variables, N = 372.

| Variables | Mean | Std. D |
|---|---|---|
| Product | 2.18 | 0.83 |
| Promotion | 2.25 | 0.87 |
| Image | 2.29 | 0.93 |
| Price | 2.82 | 0.81 |
| Physical evidence | 2.37 | 0.95 |
| Process | 2.21 | 0.75 |
| Place | 2.54 | 0.88 |
| People | 2.44 | 0.93 |
| Composite mean | 2.41 | 0.89 |
| Tourism product development (dependent variable) | 2.18 | 0.83 |

Source: Researchers Survey, 2021.

### 4.3. Relationship b/n Independent Variables and Tourism Product Development

The highest degree of association (r = 0.676, *p* < 0.001) was observed between promotion and tourism product development. The second highest degree of association (r = 0.640, *p* < 0.001) was observed between product and tourism product development. The third and positive correlation (r = 0.567, *p* < 0.001) was between distribution and tourism product development. A tourism distribution system is a way of making tourism products (goods/services) available and accessible for the targeted travelers [19]. The fourth (r = 0.523, *p* < 0.001) is between tourism organization brand image and tourism product development. The correlation analysis further revealed the existence of positive and significant associations between people and tourism product development (r = 0.472, *p* < 0.001).

Price was the only independent variable that was inversely related to tourism product development but significantly (r = −0.341, $p < 0.001$). As the price of tourism product increases the level of tourism product development decreases reversely. There were positive and less significant associations between physical evidence and tourism product development (r = 0.389, $p < 0.001$). Process positively and less significantly associated with tourism product development (r = 0.242, $p < 0.001$).

In general, the researchers concluded that all the independent variables positively (except price) and significantly correlated ($p < 0.001$) with tourism product development.

### 4.4. Regression Assumptions

The authors checked that histogram, Kurtosis, and skewness were procedures of normality expression.

The histogram graph (see Figure 2) indicates normality if it is bell-shaped from the origin [43].

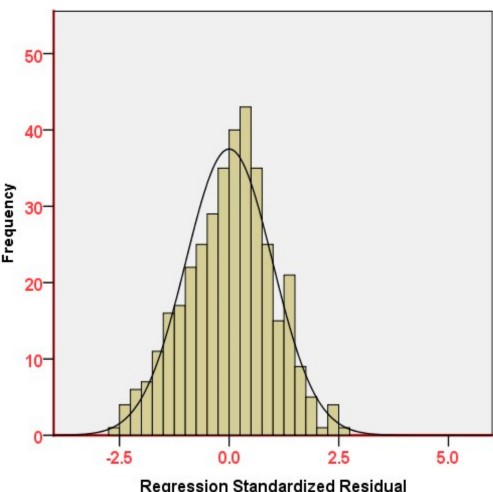

**Figure 2.** Histogram.

Multi-collinearity test happened when there is a high correlation between variables [44]. When there is high multi-collinearity among independent variables then the regression coefficient lacks uniqueness, tolerance, and variance inflation factor (VIF). If the value of VIF is less than 10 and the tolerance value is greater than 0.1, it indicates freedom from the multi-collinearity problem [43]. (See multi-collinearity diagonistics in Table 6)

**Table 6.** Multi-collinearity tests.

| | Model | Tolerance | VIF |
|---|---|---|---|
| 1 | Product | 0.677 | 1.478 |
| | Promotion | 0.729 | 1.372 |
| | Price | 0.881 | 1.136 |
| | Image | 0.804 | 1.244 |
| | People | 0.846 | 1.181 |
| | Place | 0.745 | 1.343 |
| | Process | 0.947 | 1.056 |
| | Physical evidence | 0.891 | 1.122 |

Dependent Variable: tourism product development. Source: Researcher Survey, 2021.

### 4.4.1. Regression Output and Interpretation

The multiple regression model examines the relationship of an independent variable over dependent variables [45]. In Steven's work, the regression coefficient of predictors shows the degree of their effect on the dependent variable and the total effect of all independent variables measured as a whole in ($R^2$) to the dependent variable.

### 4.4.2. Model Summary

Table 7 shows that promotion, product, image, price, place, people, physical evidence, and process had R (0.912), which indicates that they have a strong effect on the dependent variable (tourism product development). The ($R^2$) is 0.832, which is very high and interpreted as 83.2% of the variance in the dependent variable being explained by the independent variables (marketing mixes and image).

**Table 7.** Model summary.

| Model | R | R Square | Adjusted R Square | Std. Error of the Estimate | Durbin-Watson |
|---|---|---|---|---|---|
| 1 | 0.912 [a] | 0.832 | 0.829 | 1.56136 | 1.841 |

[a]: Dependent Variable: tourism product development. Source: Researchers' Survey, 2021.

### 4.4.3. ANOVA Test

From Table 8, of analysis of (ANOVA), F statistics shows how the fitness of the model is significant ($p$ = 0.000).

**Table 8.** ANOVA test.

| Model | | Sum of Squares | df | Mean Square | F | Sig. |
|---|---|---|---|---|---|---|
| | Regression | 4393.136 | 8 | 549.142 | 225.257 | 0.000 |
| 1 | Residual | 884.937 | 363 | 2.438 | | |
| | Total | 5278.073 | 371 | | | |

Source: Researcher Survey, 2021.

Table 9 shows the influence of the predictors' variables on the dependent variables. Depending on unstandardized beta coefficient analysis, the independent variables have a strong contribution to the existence of the dependent variable. This beta coefficient indicates that the average amount of change in the dependent variable is caused by a unit change in the independent variable. Therefore, the researcher used the following stated model to show that the predictors level of the determinant as follows:

$$Y = \beta 0 + \beta 1 X1 + \beta 2 X2 + \beta 3 X3 + \beta 3 X4 + \beta 3 X5 + e$$

where: Y is tourism product development, $\beta 0$ = the constant (coefficient of intercept), $\beta 1$ = regression coefficient of promotion, $\beta 2$ = regression coefficient of product $\beta 3$ = regression coefficient of image $\beta 4$ = regression coefficient of place, $\beta 5$ = regression coefficient of physical evidence, $\beta 6$ = regression coefficient process, $\beta 7$ = regression coefficient of people, $\beta 8$ = regression coefficient of price.

**Table 9.** Summarized beta coefficients of the regression model.

| Model | Unstandardized Coefficients | | Standardized Coefficients | t | Sig. |
|---|---|---|---|---|---|
| | B | Std. Error | Beta | | |
| (Constant) | −1.133 | 0.472 | | −2.403 | 0.017 |
| Product | 0.273 | 0.030 | 0.241 | 9.209 | 0.000 |
| Promotion | 0.316 | 0.022 | 0.365 | 14.488 | 0.000 |
| Price | −0.117 | 0.018 | −0.151 | −6.611 | 0.000 |
| Image | 0.221 | 0.032 | 0.163 | 6.816 | 0.000 |
| People | 0.140 | 0.018 | 0.187 | 7.989 | 0.000 |
| Place | 0.176 | 0.020 | 0.218 | 8.769 | 0.000 |
| Process | 0.143 | 0.026 | 0.123 | 5.582 | 0.000 |
| Physical evidence | 0.165 | 0.030 | 0.125 | 5.510 | 0.000 |

Source: Researchers' Survey, 2021.

Then, the above equation can be expressed in terms of b-values as follows:

Tourism product development = −1.133 + 0.316 promotion + 0.273 product + 0.221 image + 0.176 place + 0.165 physical evidence + 0.143 process + 0.140 people − 0.117 price.

The above equation can be interpreted as

Promotion: a one-unit coefficient increase in the promotion will increase tourism product development by 0.316 units. Promotion is encouraging the users for having products [25].

Product: a one-unit increase in the product will increase tourism product development by 0.273 units. Updated product design and development help for the development of tourism products [25].

Image: a one-unit coefficient increase in the image will increase tourism product development by 0.221 units. Brand image supports the maximum sales of products [25].

Distribution/place: a one-unit coefficient increase in distribution or place will increase tourism product development by0.143 units. Good relationships between providers and users decrease the extra costs [25].

Physical evidence: a one-unit increase in one unit in physical evidence will increase tourism product development by 0.143 coefficient units. Attractive and good physical evidence guaranteed the loyalty of customers at the organization [32].

Process: a one-unit increase in the process will increase tourism product development by 0.143 units. The process is the way of delivering the products to the customers [33].

People: a one-unit increase in people will increase tourism product development by 0.140 units.

Price: a one-unit increase in price will decrease tourism product development by −0.117 units. Acceptable pricing reduces the competition and increases the market share [25].

To recap, from these regression results of the seven marketing mixes, promotion was the first variable that determined tourism product development followed by product, place, physical evidence, process, people, and price respectively affecting positively (except price) tourism product development. (See Table 10 regarding the hypothesis testing results and the effect of independent variables on the outcome variable)

**Table 10.** Summary of hypothesis testing.

| Hypothesis | Results | Conclusion/Decision |
|---|---|---|
| **Hypothesis (H1):** *There is a significant relationship between product and tourism product development.* | β = 0.273; *p* = 0.000; Positive, significant | H1: supported |
| **Hypothesis (H2):** *There is a significant relationship between price and tourism product development.* | β = −0.117; *p* = 0.000 Negative, significant | H2: not supported |
| **Hypothesis (H3):** *There is a significant relationship between place (distribution) and tourism product development.* | B = 0.176; *p* = 0.000 Positive, significant | H3: supported |
| **Hypothesis (H4):** *There is a significant relationship between promotion and tourism product development.* | B = 0.316; *p* = 0.000 Positive, significant | H4: supported |
| **Hypothesis (H5):** *There is a significant relationship between people and tourism product development.* | B = 0.140; *p* = 0.000 positive, significant | H5: supported |
| **Hypothesis (H6):** *There is a significant relationship between process and tourism product development.* | β= 0.143; *p* = 0.000; positive, significant | H6: supported |
| **Hypothesis (H7):** *There is a significant relationship between physical evidence and tourism product development.* | β= 0.165; *p* = 0.000 positive, significant | H7: supported |
| **Hypothesis (H8):** *There is a significant relationship between tourism product brand image and tourism product development.* | B = 0.221; *p* = 0.000 positive, significant | H8: supported |

*4.5. Tourism Product Strategies*

4.5.1. Market Penetration Strategy

This strategy mainly focused on existing tourism products that help the destinations to get a higher share of the market. There are some techniques used for the effectiveness of these strategies. Reducing the price of tourism products to attract new customers, increasing the distribution channels and promotional activities, and understanding competitors' power are basic techniques that are used in the success of the strategy. Batu and surrounding tourism business interviewees responded as follows. Regarding price reduction, the interviewee (MMH) replied that their hotel offers price discounts during low seasons to attract tourists and to reduce the perishability of the hotel's products. The hotel sells its products at the breakeven point during the low season. Contractual agreement price offers and package sales are some of the techniques they used. The resident customers were price-sensitive and tourists largely paid without price complaint. Other interviewees (PAN, BAC, CRM, LBLM, TSS, OGG, AVM, HGM, and LVNC) answered that they need to reduce the price of the goods but the cost of a resource or raw material from suppliers challenges them to reduce the price. Therefore, there was a big challenge to reduce the price of a tourism product among the providers. This situation implied that there was no pricing strategy effectively in the study area. The researchers compared this interview with collected and analyzed questionnaire data and concluded price had inversely related with product development too.

Most interviewees responded that they would not use effective distribution channels. (PAN, BAC, CRM, LBLM, OGG, AVM, HGM, and LVNC) use online distribution channels like TripAdvisor and interviewee but interview code MMH said that:

"We have a head office in Addis Ababa which helps to facilitate and reduce travel barriers for their international tourist and facilitate domestic tourists and corporate agreed companies with our hotel. In addition to this, we have other hotels in a different part of the country which helps as a distribution channel by recommending travelers to our hotel when they arrived at Batu/Ziway"

Therefore, the researchers understood that there were not enough distribution channels used by most of the tourism businesses in the study area.

Regarding promotional activities, they were very low according to the interviewee response (MMH, PAN, BAC, CRM, LBLM, OGG, AVM, HGM, LVNC) and they did not use promotional tools. Their best promotional activity is customers' recommendation and walk-in guests is their common type of customers. Therefore, the researcher concluded that the promotional activity is very low.

In the study area, there were competitors who participated in similar tourism businesses. The interviewee (MMH) responded that

"We know our competitors well and they were finger counted even in numbers but we are the winners and for the time being we don't worry about competitors rival but we expect the competition will be increased. Therefore, we carefully follow up with future potential competitors"

Interviewees (MMH, PAN, BAC, CRM, LBLM, OGG, AVM, HGM, LVNC) think they had competitors and they understood who their competitors were. However, they did not care about their competitors because they had the same ability to compute and they were under similar computational advantage.

4.5.2. New Tourism Products Development Strategy

This strategy is used when the organization or the destination strongly understands that the currently existing market needs some modifications or else the current market would be at risk. These strategies use different techniques which support the effectiveness of strategies. It includes merging resources which are used for developing their new tourism products to cater to their existing tourism products and developing a strategic partnership with other firms to use their distribution channel and brand names for the

new tourism product development strategy. Then the interviewer asked the interviewees whether they employed new tourism product development strategies.

Interviewee TSS responded as

"We don't conduct formal research to gather the needs of customers, but we informally know what our customers need, as you know that our products are highly fashionable and somehow luxury the customer's order show as to how the needs of customers shifted and demanded some sorts of products, then we start to produce to the trend of the demand shift"

However, the interviewee (MMH) reflected that:

"We want to add and consider what our customers need. For that reason, we always find out a new product, by the way, this new product is not produced for the needs and satisfaction of our customers only but they help us in order to diversify new business stream and revenue increment"

Most of the interviewees (PAN, BAC, CRM, LBLM, OGG, AVM, HGM, LVNC) did not invest in finding new needs and research. Therefore, it can be concluded that there was very little searching of new needs of customers. If the customers cannot be offered the needs they want in the market, the concerned business could fail in the market [19].

### 4.5.3. Market Development Strategy

This strategy focused on the expansion of new markets within the existed tourism products. That means without creating or innovating the current tourism products, it is just expanding the market into new profiles of customers like expanding to new tourism geographic areas, customer segments, etc. These strategies encouraged addressing all the potential customers and ensuring the efficient utilization of existing tourism products' development. Market development strategy uses different techniques for its success. Using their own proprietary technology helps with the entry of new market, considering the acceptability of the tourism products' development by the consumer in the new market. The interviewees responded that they did not expand their market when this interview was held. They were not geographically expanding their products and creating new segmentation of customers; however, interviewee code MMH commented:

"We geographically segmented to different Ethiopian parts. Adama, Addis Ababa, Arbaminch, Hawassa were some of our destiny where our product was available and some of the rest of towns were our potential for future expansions"

In addition to code (MMH), code (TSS) also segmented their product to the profile characteristics of their customers (segmentation by age, sex, culture). Moreover, those codes (MMH) and (TSS) expand their new market based on customers' needs. According to the interview with the interviewee (TSS), they expand their market based on the characteristics of customers.

"Gender and age are the main segmentation of our product. Moreover, female customers were frequent buyers and highly fashionable in comparison of males and babies are also the main customers of our products"

## 5. Conclusions and Recommendations

### 5.1. Conclusions

Marketing mixes which are promotion, product, place, physical evidence, people, and process have significantly and positively affected the development of tourism products in the study area. However, the price was negatively and significantly determining tourism product development. The findings revealed that low promotional habits in the tourism product provision organizations lead to low development of tourism product development. Most tourism product providers are not using promotional tools effectively and customer word of mouth is the main promotional tool. Poor product design and less innovative prac-



tices negatively affect the development of tourism products. Since people's or users' wants and needs are varying constantly, products shall be designed and modified accordingly.

Distributional channels, especially the use of tour operators and travel agents, tour guides, and other distributors are performing at a very low level at the study area. Finger counted employees only take some tourism and hotel profession training and use local human powers that are unprofessional and inexperienced in most tourism organizations.

High pricing of tourism products which leads to less demand for those tourism products in the study area highly affects the tourism product development because of customer price elasticity. On the other hand, the process of service provision is deferring which can be a reason for low development of tourism product development via customer dissatisfaction.

Unstandardized buildings, physical environments of tourism organizations, and decorations have been observed in the study area. The attractiveness of the service provision center is fundamental to increasing the demand for tourism products. Besides, the brand image significantly and positively affects the development of tourism products in the study area. However, the tourism product providers could not build brand identity. Locally known tourism product providers are available in the area that can utilize tourism resources which could help them to be competent and build the brand image in the future if worked on.

Most tourism business organizations were not geographically expanding their products and creating new segmentation of customers. Additionally, the tourism business did not expand their goods and services to different neighboring cities. They are unable to expand their product to deliver in other geographical areas because of financial constraints.

The market penetrating strategy was not applied well in the study areas of tourism products. There was high tourism product price, either no use or single distribution, low promotional activities, and less understanding of competitors among tourism business.

Diversified services besides the tourism services like transportation or shuttle services, banking services, exchange rate services, mobile recharge vouchers, and outdoor and indoor laundry services existed but were very few in number. Parks start to offer tourism-related products like food and beverage service besides their main products. Besides, tourism product development can be innovated through research development which helps to cater their existing tourism products and developing strategic partnerships with other firms in order to use their distribution channel and brands at a low level.

*5.2. Recommendations*

Based on the findings of the study, tourism product development is at a low level. It needs strong collaboration of government, residents, investors, and private sectors concerning tourism industry bodies in the area.

There are tourism resources in the study area, but they have not been utilized and developed into tourism products. That is better if the government creates an opportunity for the potentiated investors by encouraging through facilitating all the processes of investment and incentives such as being free from tax exemption for some periods. In addition to this, tourism business should produce goods and services (unique and quality products) and should get experience from other destinations.

Tourism offices should employ different promotional tools like advertisement on mainstream media, online website development, and use of social media to attract actual and prospective tourists and investors in the tourism industry.

Today, the price of tourism products has become highly inflated. Therefore, hotels, resorts, guest houses, and restaurants should have reduced their cost by using local resources (locally produced crops, fruits, vegetables).

Attention should be paid to professional, qualified, and skilled human power by tourism business. The way of serving, the standard time of service delivery, punctuality, and integrity and courteousness has a great influence on the mind of users to either retain or attract them.

Being innovative or creating newly adopted tourism products and diversifying products is advisable to hotels and lodges particularly focusing on local (cultural) products, which could bring high competitive advantage even over nearby destinations.

**Author Contributions:** Conceptualization, T.H.B.; Writing—review & editing, K.B.M. All authors have read and agreed to the published version of the manuscript.

**Funding:** This research received no external funding.

**Institutional Review Board Statement:** The study was conducted according to the research guidelines of Hawassa University, Ethiopia, and approved by the research committee on 2 August 2021.

**Informed Consent Statement:** Informed consent was obtained from all subjects involved in the study.

**Data Availability Statement:** Not Applicable.

**Acknowledgments:** First and foremost, we thank almighty God for giving us the courage and strength to complete this study. Secondly, this work was not the result of an independent effort. It was accomplished with the assistance of many people to whom the authors are very indebted. Our heartfelt gratefulness goes to participants of the study for their willingness, enthusiasm, thoughts, and for providing essential information pertaining to the issue under investigation. We also would like to express our genuine thanks to Hawassa University Tourism Management department staff, anonymous reviewers of the manuscript and managing editors of Sustainability (the assistant editor of the journal: Bianca Bode, and the academic editors: Dimitrios Aidonis, Naoum Tsolakis and Charisios Achillas) for their constructive comments.

**Conflicts of Interest:** The authors declare no conflict of interest.

## Appendix A. Questionnaire

Part 1: Indicators of tourism product development
Dear respondents, please rate the existing tourism product in Batu and its vicinity.
**1** = Strongly disagree; **2** = Disagree; **3** = Neutral; **4** = Agree; **5** = strongly agree

| No. | Tourism Product | 1 | 2 | 3 | 4 | 5 |
|---|---|---|---|---|---|---|
| 1 | Batu city and its surroundings have well conserved and protected natural tourism resources (forests, parks, birds, water bodies, animals, and plants) | | | | | |
| 2 | There are well visited and managed cultural or human-made tourism resources (buildings, obelisks, museums, palaces, arts) | | | | | |
| 3 | Batu and surroundings are suitable for hosting event tourism (sports, conferences, meetings, exhibitions, etc.) | | | | | |
| 4 | There are modern accommodations (hotels, pensions, guest houses, lodges, etc.) in the city and its surroundings | | | | | |
| 5 | There are qualified food and beverage service providers (restaurants, lounges, hotels, cultural food houses, bars) | | | | | |

Part 2. Marketing mix determinants of tourism product development
Dear respondent, please rate the determinants of tourism product development
**1** = Strongly disagree; **2** = Disagree; **3** = Neutral; **4** = Agree; **5** = strongly agree

| No. | New Tourism Product Development | 1 | 2 | 3 | 4 | 5 |
|---|---|---|---|---|---|---|
| 1 | The organization develops new products for the market | | | | | |
| 2 | The organization improves its product | | | | | |
| 3 | The organization provides locally used products | | | | | |
| 4 | The organization uses the latest design for its product | | | | | |
| Price | | | | | | |

| 1 | The organization offers discounts to its customers |
|---|---|
| 2 | Customers are happy with the price charged |
| 3 | The organization gives credit service for customers |
| 4 | The organization has the power of changing prices in the market |

**Place**

| 1 | The organization uses intermediaries |
|---|---|
| 2 | The organization uses its best alternative channel |
| 3 | The organization reduces the perishability of products |
| 4 | The organization is easily accessible for customers |
| 5 | The organization is at a convenient place to get other services |

**People**

| 1 | The organization gives training for its employees periodically |
|---|---|
| 2 | The organization recruits the right employees for the position |
| 3 | The organization's employees are certified |
| 4 | The working environment of the organization is safe |
| 5 | The organization's employees are happy with their salary |

**Promotion**

| 1 | The organization advertises its product by using media |
|---|---|
| 2 | The organization sells products to customers face to face |
| 3 | The organization's product is publicized |
| 4 | The organization gives additional free services to the customers |
| 5 | The organization's customers have positive feedback for the service they receive |

**Physical evidence**

| 1 | The organization's surrounding environment is attractive |
|---|---|
| 2 | The design of the organization's building is new and attractive |
| 3 | The organization's product delivery area is well decorated |

**Process**

| 1 | The organization delivers the same standard products all the time |
|---|---|
| 2 | The organization delivers its product within a short period |
| 3 | The organization saves customers unnecessary payments incurred in the process of receiving the product |

Part 3. Branding image determination on tourism product development

**Brand Image**

| 1 | The organization has better experience in the industry |
|---|---|
| 2 | The organization's product is well known by its customers |
| 3 | The organization is the customers' first choice |

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
