# Peer review of "Determinants of Tourism Product Development in Southeast Ethiopia: Marketing Perspectives"

_sustainability, doi:10.3390/su132313263_

Round 1

Reviewer 1 Report

In general, I consider that the paper presented is of interest to the scientific field addressed. It is especially interesting because of the geographical area where the research is carried out, where there are no similar studies.
I consider that the structure is adequate and all the parts are correctly explained. 
For the publication of this research work it would be necessary to revise the way of citing in the text and the bibliographical references, since they do not meet the technical requirements of the journal.

Author Response

Question: In-text citations and references must be revised and meet the technical requirements of the journal.

Authors’ response: All the in-text citations and reference section is revised as per APA 7th edition referencing and citation format. Besides, the reviewer affirmed that the manuscript is sufficient for publication.  

Regards!

Reviewer 2 Report

The manuscript ‘ Determinants of tourism product development in Southeast Ethiopia: Marketing perspectives’ discusses an important and up-to-date topic, relevant for many tourism destinations. However, the so called Quality of Presentation has to be improved significantly, as in its present form the manuscript lacks several important elements of a scientific paper and has several unnecessary ones. There are also some other elements that need to be either added or upgraded at least. Consequently in its present form the manuscript does not provide all the necessary information on the research, while explaining at some points matters that are well known.

The most important things to improve are:

  1. The abstract needs revision. It should be more concise.
  2. Section 3 Methods and materials, especially 3.2 Research design, is very general. It does not include any precise information on the data categories describing selected to describe independent variables as Product, Process etc. All we know is that x1= product, ok but what exactly has been reflected in this category, how it has been measured, what units have been applied in case of quantitative data, etc. So, at least a table explaining this issue should be added.
  3. The method does not really explain the survey clearly and sufficiently, leaving the reader with some unanswered questions, e.g. where, when and how was the data collected? What questions were included? What was the response rate? And many other ... So, the information on the survey should be collected and amended in one section.
  4. On the other hand, explaining what the classical components of regression mean is not necessary e.g. lines 441-444. Putting just the results with references to the literature is sufficient.
  5. As very important information is missing now, evaluation of the following parts will be possible when the manuscript is improved as listed above.
  6. English and edition should be significantly improved.

Concluding, the manuscript needs revision.

Author Response

Reviewer’s Question:

The reviewer emphasised that the manuscript has to be revised. The following points have been raised.

  1. The abstract needs revision. It should be more concise.
  2. Section 3 Methods and materials, especially 3.2 Research design, is very general. It does not include any precise information on the data categories describing selected to describe independent variables as Product, Process etc. All we know is that x1= product, ok but what exactly has been reflected in this category, how it has been measured, what units have been applied in case of quantitative data, etc. So, at least a table explaining this issue should be added.
  3. The method does not really explain the survey clearly and sufficiently, leaving the reader with some unanswered questions, e.g. where, when and how was the data collected? What questions were included? What was the response rate? And many other ... So, the information on the survey should be collected and amended in one section.
  4. On the other hand, explaining what the classical components of regression mean is not necessary e.g. lines 441-444. Putting just the results with references to the literature is sufficient.
  5. As very important information is missing now, evaluation of the following parts will be possible when the manuscript is improved as listed above.
  6. English and edition should be significantly improved.

Authors’ Response

  1. The abstract is revised and presented as follows.

Tourism has been given much attention in developing countries like Ethiopia. In this regard, tourism product development played a great role to achieve sustainable developmental goals. The study aims to examine the determinants of tourism product development in southeast Ethiopia. The research employed a mixed research approach, and descriptive and explanatory research design was used. Both secondary and primary data sources were in place to obtain the relevant data. A total of 398 samples were employed to collect the data. A convenience sampling technique was employed to select domestic tourists, walk-in guests and purposive sampling was applied to select marketing managers of hotels, lodges, boat associations, park administration, resorts, restaurants, nightclubs, and guest houses. The qualitative data was analyzed through thematic analysis and the quantitative data was analyzed using descriptive and inferential statistics by computing SPSS. The study revealed that tourism product development is affected by marketing mixes of which promotion was the major factor of tourism product development and price was the only tool-related inversely with tourism product development. The brand image also positively determined tourism product development. The results showed that the exercise of tourism product development strategies among tourism businesses was low. The main implication drawn from the study is that the tourism businesses have to develop new customer profiles or segmenting in customers in their specified characteristics like gender, level of income, and age. And the need to invest in research and development of the current market to develop new tourism products is of great importance.

Keywords: Tourism, Ethiopia, Product development, Marketing mixes, brand image, determinant, Sustainability 

  1. Research Design (Question 2 & 3)

Authors’ response:

After revision, the research design is specific and designed to address the objectives of the research. Variables are clearly stated. Kindly see the following revisions.

In this study, descriptive and explanatory research designs were employed. The descriptive research design was used for describing the existing situation or phenomenon of the study in detail (Kothari, 2004). Therefore, this research design was employed to examine the strategies used by tourism businesses in the study area to develop tourism products.

The explanatory research method is applied to examine the cause and effect relationship between variables (Bhattacherje, 2012). Therefore, the current study employed an explanatory research design to explain and analyze the effect of marketing mixes on tourism products development. The independent variables are existing tourism product, price, place, promotion, people, physical evidence, process and brand image. However, the dependent variable is new tourism product development.

A qualitative approach was used to narrate and interpret qualitative responses from interviews respondents. A quantitative approach was in place to generalize the findings resulting from descriptive and inferential statistics.

The data was collected in Batu town and surrounding tourism product providers. Primary data and secondary data were employed from September 2020 to July 2021 through primary data (questionnaire and interview). The total questionnaires that have been distributed were 384 of which 372 (96.8 % response rate) were valid for analysis. The items measuring the dependent and independent variables are attached in the appendix part.

Specifically, section 3.6 Variables and Model specification and Table 1 depicts the dependent and independent variables and related issues that have to be addressed.

Question 4 (less significant portions of descriptions: lines 441-444) to be omitted.

Response: We remove the parts (lines 441-444) from the manuscript as they have little significance.

In general, the authors have carried out the necessary modifications (revisions and editions) as per the comments of the reviewers.

Regards!

Reviewer 3 Report

The idea and research are interesting. The study has potential to be a good article.

The structure of the article is chaotic. Please, organize the research results (maybe separately the results of the survey and interviews, or in some other way).

All tables numbering and font sizes must be ordered.

Poorly described study desing and methods of interview. How was the interview questionnaire carried out? The interview research results are chaotic. I am asking for a synthesis of the results of qualitative research.

The conclusions are very limited, please elaborate. Please devote a paragraph to the discussion. Which research results are consistent with previous research and what are not?

The article has a lot of technical errors: line 21, 57, 74, 116, 160, 162, 167, 182, 248, 312 and other. Lines 335-339 formulas should be corrected as in guides for authors.

Table 3 is unclear: different fonts and font sizes - need to be organized, especially headings. And the table should be on one page integrally.

Author Response

Reviewer’s comments and questions

  1. The idea and research are interesting. The study has the potential to be a good article.
  2. The structure of the article is chaotic. Please, organize the research results (maybe separately the results of the survey and interviews, or in some other way).
  3. All table numbering and font sizes must be ordered.
  4. Poorly described study design and methods of interview. How was the interview questionnaire carried out? The interview research results are chaotic. I am asking for a synthesis of the results of qualitative research.
  5. The conclusions are very limited, please elaborate. Please devote a paragraph to the discussion. Which research results are consistent with previous research and what are not?
  6. The article has a lot of technical errors: line 21, 57, 74, 116, 160, 162, 167, 182, 248, 312 and others. Lines 335-339 formulas should be corrected as in guides for authors.
  7. Table 3 is unclear: different fonts and font sizes - need to be organized, especially headings. And the table should be on one page integrally.

Responses:

  1. The results of the study are separately presented. the survey results are demonstrated from page 16 to 24 (descriptive and inferential statistics outputs and discussions are elaborated separately). And the qualitative findings i.e explaining tourism product development strategies are discussed particularly under section “Tourism product strategies” (page 24- 27).
  2. Font size and table numbering are rearranged.
  3. The qualitative data was gathered through undertaking interviews in line with tourism product development strategies applied by tourism businesses. The interview checklist was prepared based on Ansoff Matrix on Tourism Product development strategies. The interviewees were purposely selected from tourism product providers and the interview was carried out until the attainment of data saturation. The data saturation concept recommends that the point where the researchers get similar answers from the interviewees could lead to stopping gathering further responses on the issue (Saunders, Sim, Kingston, Baker, & Waterfield, 2017). Accordingly, the interview data was saturated at the point of the 10th interviewee in this study. Therefore, the researchers select mostly available tourism business organizations for the interview (marketing manager, general manager, coordinators, etc.) of resort hotels, national parks, business associations, cultural restaurants, beach & lodge, souvenir shop, guest house, tourist hotel, night clubs. Generally, the study employed 10 respondents to undertake the interviews.

  1. The conclusion is revised and elaborated as follows.

Marketing mixes which are promotion, product, place, physical evidence, people and process have significantly and positively affected the development of tourism products in the study area. However, the price was negatively and significantly determining tourism product development. The findings revealed that low promotional habits in the tourism product provision organizations lead low development of tourism product development. Most tourism product providers are not used promotional tools effectively and customer word of mouth is the main promotional tool. Poor product design and less innovative practices negatively affect the development of tourism products. Since people or users' wants and needs are varying instantly, products shall be designed and modified accordingly.

Distributional channels especially the use of tour operators and travel agents, tour guide and other distributors are performing at a very low level in the study area. Finger counted employees only take some tourism and hotel profession training and use local human powers that are unprofessional and inexperienced in most tourism organizations.

High pricing of tourism products which leads to less demand for those tourism products in the study area, highly affects the tourism product development because of customer price elasticity. On the other hand, the process of service provision is deferring which can be a reason for low development of tourism product development via customer dissatisfaction.

Unstandardized buildings, physical environments of tourism organizations and decorations have seen observed in the study area. The attractiveness of the service provision center is fundamental to increasing the demand for tourism products. Besides, the brand image significantly and positively affects the development of tourism products in the study area. However, the tourism product providers couldn’t build brand identity. Locally known tourism product providers are available in the area that can utilize tourism resources which could help them to be competent and built a brand image in the future if worked on. 

Most tourism business organizations were not geographically expanding their products and create new segmentation of customers. Additionally, the tourism business did not expand their goods and services to different neighbor cities. They are unable to expand their product to deliver in other geographical areas because of financial constraints.       

The market penetrating strategy was not applied well in the study areas of tourism products. There was high tourism product price, either no use or single distribution, low promotional activities and less understanding of competitors among tourism business. Diversified services besides the tourism services like transportation or shuttle services, banking services, exchange rate services, mobile recharge vouchers, and outdoor and indoor laundry services were existed but very little in number. Parks start to offer tourism-related products like food and beverage service besides their main products. Besides, tourism product development can be innovated through research development which helps to cater their existing tourism products and developing strategic partnerships with other firms in order to use their distribution channel and brands at a low level.

  1. Technical errors specified by the reviewer and other faults have been critically solved.

Formula

  1. to determine sample size: ,Where  n= is the sample size; Z2 =1.96 is the desired confidence level for 95%; P = 0.5 is the estimated proportion of an attribute that is present in the population; E = 0.05 is the desired level of precision, 5%.  Hence, n = (1.96)2(0.5) (0.5)/ (0.05)2 =
  2.  

Y = dependent variable

X1, X2, X3 = independent variables, 

β0 = Y intercept (constant term)

β1, β2, β3 … = Slop coefficient for each independent variables

έ = the model error 

Tables: Tables are appropriately numbered, font size properly and consistently used and corrected.

The Researchers have carried out the necessary modifications given by the reviewer.

Regards!

Round 2

Reviewer 2 Report

The revised manuscript discusses the topic and presents all elements of research in an improved form, meeting the criteria of a scientific paper.

Reviewer 3 Report

Pay attention to technical errors such as spacing